# DNA Methylation Profiling for Diagnosing Undifferentiated Sarcoma with Capicua Transcriptional Receptor (*CIC*) Alterations

**DOI:** 10.3390/ijms21051818

**Published:** 2020-03-06

**Authors:** Evelina Miele, Rita De Vito, Andrea Ciolfi, Lucia Pedace, Ida Russo, Maria Debora De Pasquale, Angela Di Giannatale, Alessandro Crocoli, Biagio De Angelis, Marco Tartaglia, Rita Alaggio, Giuseppe Maria Milano

**Affiliations:** 1Department of Pediatric Onco-Hematology and Cell and Gene Therapy, IRCCS Bambino Gesù Children’s Hospital, 00165 Rome, Italy; lucia.pedace@opbg.net (L.P.); ida.russo@opbg.net (I.R.); mdebora.depasquale@opbg.net (M.D.D.P.); angela.digiannatale@opbg.net (A.D.G.); biagio.deangelis@opbg.net (B.D.A.); giuseppemaria.milano@opbg.net (G.M.M.); 2Department of Laboratories, Pathology Unit, IRCCS Bambino Gesù Children’s Hospital, 00165 Rome, Italy; rita.devito@opbg.net (R.D.V.); rita.alaggio@opbg.net (R.A.); 3Genetics and Rare Diseases Research Division, Bambino Gesù Children’s Hospital (IRCCS), 00165 Rome, Italy; andrea.ciolfi@opbg.net (A.C.); marco.tartaglia@opbg.net (M.T.); 4Department of Surgery, IRCCS Bambino Gesù Children’s Hospital, 00165 Rome, Italy; alessandro.crocoli@opbg.net

**Keywords:** undifferentiated sarcoma, DNA methylation profiling, *CIC* alteration, diagnosis

## Abstract

Undifferentiated soft tissue sarcomas are a group of diagnostically challenging tumors in the pediatric population. Molecular techniques are instrumental for the categorization and differential diagnosis of these tumors. A subgroup of recently identified soft tissue sarcomas with undifferentiated round cell morphology was characterized by Capicua transcriptional receptor (*CIC*) rearrangements. Recently, an array-based DNA methylation analysis of undifferentiated tumors with small blue round cell histology was shown to provide a highly robust and reproducible approach for precisely classifying this diagnostically challenging group of tumors. We describe the case of an undifferentiated sarcoma of the abdominal wall in a 12-year-old girl. The patient presented with a voluminous mass of the abdominal wall, and multiple micro-nodules in the right lung. The tumor was unclassifiable with current immunohistochemical and molecular approaches. However, DNA methylation profiling allowed us to classify this neoplasia as small blue round cell tumor with *CIC* alterations. The patient was treated with neoadjuvant chemotherapy followed by complete surgical resection and adjuvant chemotherapy. After 22 months, the patient is disease-free and in good clinical condition. To put our experience in context, we conducted a literature review, analyzing current knowledge and state-of-the-art diagnosis, prognosis, and clinical management of *CIC* rearranged sarcomas. Our findings further support the use of DNA methylation profiling as an important tool to improve diagnosis of non-Ewing small round cell tumors.

## 1. Introduction

Primitive small blue round cell tumors (SBRCTs) in children and young adults pose a diagnostic challenge. Ewing sarcoma (ES) is the prototype for an undifferentiated sarcoma with such a phenotype, although it is shared by several other sarcoma entities together with the membrane expression of CD99 [1,2]. An advance in the diagnosis of Ewing’s sarcoma was the discovery of recurrent, highly-specific balanced translocations leading to a chimeric gene fusion involving the RNA-binding TET (translocated in liposarcoma/Ewing sarcoma breakpoint region 1/TATA box binding protein-associated factor) gene family members, mainly *EWSR1*, and members of the E26 transformation-specific (*ETS*) gene family. *EWSR1–FLI1* or *EWSR1–ERG* are the most common translocations in ES and are observed in ~85%–90% and 5%–10% of all cases, respectively [3,4]. Far rarer are other gene fusions involving different *TET* (*FUS*) and *ETS* family members (*ETV1/4*, *FEV*, and *E1A-F*) and rearrangements of *EWSR1* with non*-ETS* family genes (including *NFATc2*, *PATZ1*, *SMARCA5*, and *SP3*), which occur in fewer than 1% of ES [5,6,7,8,9,10].

Molecular techniques, including extended Fluorescent in situ hybridization (FISH) analysis and next generation sequencing, have allowed the separation of ES with canonical translocations from a subset of lesions that remains unclassified. These are termed “Ewing-like sarcomas”, or SBRCTs not otherwise specified in the last World Health Organization classification [11,12]. Although a number of SBRCTs remains unclassified, emerging molecular evidence, most notably transcriptome analysis, has helped to distinguish previously unrecognized tumor entities that many investigators now consider as distinct from ES [2,13]. These are Ewing-like sarcomas most commonly harboring *CIC* and *BCL6 Corepressor* (*BCOR*) rearrangements [1,2,14,15,16,17]. *CIC* rearranged sarcomas arise more often in soft tissues rather than in bone, show a myxoid matrix, have prominent nucleoli, and are characterized by a poor prognosis, especially for patients with metastatic or recurrent disease [1,2].

Diagnosis of such tumors is difficult due to their rarity and unspecific histological features but is possible by combining morphological, immunohistochemical, and molecular techniques [1,2]. However, FISH and total RNA sequencing may sometimes have pitfalls in identifying some rearrangements. Koelsche and colleagues recently introduced array-based DNA methylation profiling as a method with extraordinary power for clarifying the diagnoses of a cohort of tumors initially deemed SRBCT not otherwise specified [12].

Here we report a challenging diagnosis of undifferentiated sarcoma with *CIC* alteration identified with DNA methylation profiling. To further inform management of such cases, we review the available literature and discuss the current knowledge on molecular and clinical features of *CIC* rearranged sarcomas. We conclude that DNA-methylation-based tumor classification can be an important tool in advancing the diagnosis, pathological analysis, and management of these tumors.

## 2. Case Report

A 12-year-old female was referred to our hospital emergency department with recent-onset abdominal pain. No other symptoms were recorded; she had no fever, vomiting or diarrhea. Diuresis and menstrual cycle were regular as well. Clinical examination revealed a hard-elastic swelling in the meso-hypogastric region. Complete abdomen ultrasound and magnetic resonance imaging (MRI) showed a voluminous bilobed mass of approximately 13.5 cm × 7 cm, with regular margins, in the left paramedian of the anterior abdominal wall, extending from the umbilical region to the supra-bladder area and, posteriorly reaching the spinal vertebral bodies at the lumbar-sacral level. The cranial portion of the mass appeared capsulated with signal over-intensity in all impulse sequences in relation to myxoid component and high cellularity in diffusion weighted imaging (DWI). The caudal portion appeared more inhomogeneous, predominantly hypo-intense in all sequences, as for bleeding (Figure 1A–G).

An incisional biopsy of the tumor was performed. Histological examination showed proliferation of medium-sized elements, with sharp cellular contours, dimly eosinophilic cytoplasm, and polymorphic and polymetric nuclei, with evident nucleoli and atypical mitotic figures (Figure 2A–C). The cells appeared immersed in amorphous, dimly eosinophilic proteinaceous material. Hemorrhagic extravasation in the interstitium was observed. Immunohistochemical characterization was as follows: Vimentin: positive; S100: rare positive cells; CD99 (Figure 2D) and CD31: weak positivity; Neuron Specific Enolase (NSE): weak positivity; Cytokeratin (CK) MNF116: rare positive cells;CKAE1-AE3: rare positive cells; CD117: weak positivity; Friend leukemia integration 1 transcription factor (FLI1): positive; Wilms’ tumor (WT1): positive (Figure 2F), nuclear; INI: positive; Bcl6: weak positivity; Transducin-like enhancer of split 1 (TLE1): positive Epithelial membrane antigen(EMA), Synaptophysin, Actins, CK7, Desmin, Myogenin, MyoD, Leukocyte common antigen (LCA), CD30, Activin receptor-like kinase (ALK1), Myeloperoxidase (MPO), Terminal deoxynucleotidyl transferase (TdT), Melanoma-associated antigen recognized by T cells (MART1), CD34, p63, NUT, CD56, Placental alkaline phosphatase (PLAP) were all negative. The proliferation index evaluated through Ki-67 assay was high (Figure 2E).

Molecular evaluation by both FISH and PCR found no evidence of the translocation of *EWS*. PCR testing for *CIC-DUX4*, *BCOR-CCNB3*, and for the myxoid chondrosarcoma transcript was also negative. Histopathological diagnosis was determined through examination of the specimen at the Italian pediatric sarcoma central pathology panel whose conclusion was a malignant mesenchymal neoplasm/unclassifiable sarcoma.

To complete the diagnostic work-up, the patient underwent bone scintigraphy, which was negative for areas of abnormal accumulation. A Positron Emission Tomography (PET)/CT scan confirmed the presence of the mass, which was metabolically active (maximum standardized uptake value -SUV 7 and diameter 14 × 8.2 × 7.7 cm) at the left rectus abdominis muscle, with extensive development in the meso- and hypo-gastric site. Compressive phenomena in some loops of the small bowel and on the left psoas muscle were also observed. The tumor was capsulated without infiltration of the surrounding structures. There were no other areas of tracer uptake in the peritoneum, lymph nodes, lungs, and skeletal segments. A CT scan showed millimetric nodules in the latero-basal segment of the lower lobe of the right lung, in the dorsal segment of the upper lobe of the right lung in the sub-pleural area, and in the middle lobe, where metabolic activity could not be assessed due to the small size of the nodules.

DNA methylation profiling was performed according to protocols approved by the institutional review board, after obtaining written consent from the patient’s parents. Tumor areas with the highest tumor cell content (≥70%) were selected for DNA extraction. Samples were analyzed using Illumina Infinium Human Methylation EPIC Bead Chip (EPIC) (Illumina, San Diego, CA, USA) arrays according to the manufacturer’s instructions, as previously reported [18,19,20]. In detail, 500 ng DNA were used as input material for fresh frozen tissue. Generated methylation data were compared with the Heidelberg sarcoma classifier [21] to assign a subgroup score for the tumor compared to 63 different sarcoma methylation classes identified to date.

The tumor had a score of 0.98 in the “methylation class SRBCT with *CIC* alteration”. Global profiling methylation data were also compared to 32 samples randomly extracted from internal and external datasets [21] among those classifying as central nervous system neuroblastoma *FOXR2* (*CNS-NB-FOXR2*), *CIC* rearranged sarcoma (*EFT-CIC*), high-grade neuroepithelial tumor *MN1* (*HGNET-MN1*), and high-grade neuroepithelial tumor *BCOR* (*HGNET-BCOR*) using the Heidelberg brain tumor and sarcoma classifier. Bead Chip data were analyzed by means of R (V. 3.4.4) package minfi (V. 1.24.0) to obtain normalized beta values and to perform multidimensional scaling (MDS) analysis, as previously described [18,19,20]. Our patient displayed global methylation levels close to those of Ewing Family Tumor (EFT)-*CIC*, as evidenced by MDS performed on the 1000 most variable islands in the cohort (Figure 3). The copy number variation plot showed a segmental loss on chromosome arm 1p, a deletion in 4q, and a duplication of 20q (Figure 4). These results were also confirmed by chromosomal microarray analysis, which showed the following three somatically acquired defects: 40.1 Mb deletion in 1p36.33p34.2; 1.3 microdeletion Mb in 4q35.2; 1.1 Mb microduplication in 20q13.2.

In light of the histological, molecular, and clinical findings, a neoadjuvant chemotherapy was proposed to the patient’s family. After written informed consent was obtained, neoadjuvant chemotherapy was started according to European Pediatric Soft Tissue Sarcoma Study Group (EpSSG) 2005 non-Rhabdomyosarcoma protocol (Intergroup Rhabdomyosarcoma Study-IRS stage III). After three cycles of ifosfamide (3 gr/m^2^/day for three days) and doxorubicin (37.5 mg/m^2^/day for 2 days) every three weeks, disease reevaluation showed an excellent response, with shrinkage of tumor volume (8.1 × 5.6 × 3.9 cm) (Figure 1H,I). The pulmonary nodules remained stable.

A gross total resection of the residual tumor was performed. Histological examination showed a reactive fibrous pseudocapsule surrounding the mass peripherally. Regression areas, alternating with vital areas were observed. The regression areas, which were quantified in about 60% of the total mass, showed evidence of necrosis, hemorrhagic infarction, and histiocytic infiltration by elements with a foamy and/or pigmented aspect due to the presence of abundant hemosiderin in the interstice. The vital areas were irregularly distributed within the mass and quantified in about 40% of the total. They consisted of elements with a morphology and immunohistochemical pattern substantially similar to the one detected in the diagnostic biopsy. The margins of resection were negative.

Three cycles of adjuvant chemotherapy were administered: two of ifosfamide (3 gr/m^2^/day for two days), one of ifosfamide (3 gr/m^2^/day for three days), and doxorubicin (37.5 mg/m^2^/day for 2 days) every three weeks. At the 22-month follow up, the patient was in good clinical conditions and disease free.

While the revision of this manuscript was ongoing, we further analyzed the case by next generation sequencing (Archer FusionPlex Sarcoma kit -ArcherDX, Boulder, CO, USA) and finally detected the CIC-DUX4 rearrangement: CIC (Exon20)→DUX4 (Exon1), confirming the results obtained by methylation profiling (Appendix A). Total RNA was extracted from 5 to 10 µm formalin-fixed paraffin-embedded (FFPE) tissue sections using the Qiagen miRNeasy FFPE kit (Qiagen, Valencia, CA, USA), quantified by using Qubit fluorimeter (Thermo Fisher Scientific, Waltham, MA, USA) and quality checked on the 2100 Bioanalyzer by using the RNA 6000 Nano kit (Agilent Technologies, Santa Clara, CA, USA). Anchored multiplex PCR-based libraries were prepared and run on Illumina (Miseq, San Diego, CA, USA) platforms according to the manufacturer’s instructions. The Archer Analysis bioinformatics platform was exploited for the analysis of the Archer FusionPlex panel results.

## 3. Discussion

*CIC* rearranged sarcomas are rare but represent the most frequent molecular subtype among Ewing-like undifferentiated SBRCTs [1,22], accounting for approximately 70% of SBRCTs lacking *EWSR* and *FUS* rearrangements [2,16,23,24,25]. *CIC* rearranged tumors are characterized by fusions involving *CIC*, a human homolog of *Drosophila* Capicua located on chromosome 19q13.2 and encoding a transcriptional repressor, functioning downstream of tyrosine kinase receptor signaling.

The first case of a *CIC* rearranged round cell sarcoma with t(4;19)(q35;q13.1) translocation was reported in 1996 in a 12-year-old male with an ankle mass and pulmonary metastases [26,27]. In 2006, Kawamura-Saito et al. [14] identified the molecular feature characterizing these tumors—the CIC-DUX4 (double-homeobox 4) chimeric protein resulting from this translocation in a report of two adult patients. The CIC-DUX4 fusion most often results from either t(4;19(q35;q13) or, less frequently, from t(10;19)(q26;q13), the latter involving the *DUX4* paralog *DUX4L*.

Both *DUX4* and *DUX4L* genes contain a 3.3-kb tandem repeat sequence located in the sub-telomeric regions of the long arms of chromosomes 4 and 10, respectively [28]. They encode for double-homeobox transcription factor typically expressed in germ cells as well as in the human testis, but epigenetically silenced by methylation in differentiated cells [1,29,30,31]. No significant clinical-pathologic differences have been reported between tumors harboring the t(4;19) versus the t(10;19) [31]. The CIC–DUX4 chimeric protein generally shows a largely preserved CIC domain, including the high mobility group box, fused in-frame with the C-terminus of *DUX4/L*, which loses most of its sequence in the fusion [32]. Such genetic rearrangement increases the transcriptional activity of *CIC.* The deriving chimeric protein is an oncogenic transcription factor that dysregulates the expression of its downstream targets [16]. In particular, the biological consequence is an upregulated expression of genes normally repressed by CIC, including polyoma enhancer activator 3 (*PEA3*) family members of *ETS*-related transcription factors *ETV1*, *ETV4*, and *ETV5* [14,16,24,33]. Of note, *PEA3* belongs to the family of *ETS*-related transcription factors reported as rare fusion partners of *EWSR1* in ES [22,34,35].

Even if *DUX4* is the most common partner gene of *CIC* [16,22], SBRCTs with alternate fusion partners of *CIC* have also been described. These include *FOXO4* [36,37] and *NUT* midline carcinoma family member 1 (*NUTM1*) [38] and occasionally other still-uncharacterized partner genes [25,31,36,37]. This variability of the *CIC* gene fusions has prompted the use of the term “*CIC* rearranged family of tumors” [25] to include alternate fusion partners under the definition of the entity [2]. Indeed, gene expression profiles of these tumors cluster tightly with *CIC-DUX4* and *CIC-DUX4L* rearranged tumors [25,37]. Conversely, transcriptomes of *CIC-DUX4* fusion and other *CIC* rearranged sarcomas compared with ES family of tumors (*ESFT*s) with *EWSR1-ETS* fusions show little overlap in differentially expressed genes, supporting the distinction of these tumors from ESFTs [2].

Further evidence to support *CIC* rearranged sarcomas as a distinct entity was seen in experiments conducted by Yoshimoto et al. [33]. The authors generated a mouse model expressing the human *CIC–DUX4* fusion gene (mCDS) and compared the gene expression profile of the mCDS mutant mice with that of mice with classic ES (mES). They found a large number (1661) of differentially expressed genes in mCDS or mES, suggesting that *CIC–DUX4* sarcomas are distinct from ES. Moreover, *CIC* rearranged sarcomas have been reported to show specific epigenetic signature (as discussed below) [12].

From a clinical point of view, *CIC* rearranged SRBCTs affect a wide age range (6–81 years), but are diagnosed most frequently in young adults (mean age 30 years, slightly more predominant among males) [16,31,39]. Approximately 90% of these tumors arise in soft tissues, evenly divided between trunk/pelvis and extremities and only rarely in visceral organs (around 10%) or bone (<5%) [2,16,31,39]. There are no known associated risk factors [2].

Although there are limited clinical data to date, these tumors seem to have a significantly more aggressive course when compared with classic ES [16,31,33]. In particular, in the largest series reported to date, Antonescu et al. compared the clinical follow-up of 45 patients affected by *CIC* rearranged sarcoma with localized disease with a control group of 45 ES cases, showing a 43% 5-year survival versus 77% of ES [31]. Smaller series reported similar results [25,40,41] with a median survival less than two years, despite a multimodal therapeutic approach (surgery and/or chemotherapy/and or radiotherapy) [41]. Indeed, contrary to ES, where the traditional therapeutic strategy is the neo-adjuvant chemotherapy due to the strong tumor chemo-responsiveness, responses to neoadjuvant chemotherapy in most *CIC* rearranged sarcoma patients were minimal or generally temporary with rapid onset of drug resistance and disease progression [25,40,41]. In contrast to these findings, our patient showed a dramatic response to neoadjuvant chemotherapy and, although follow-up is limited, remains disease free. Most of the data available on chemotherapy response in *CIC* rearranged sarcoma patients refer to the older population, so the discrepancy between our patient response and the literature report could be found in the young age of our patient.

The optimal therapeutic approach for *CIC* rearranged sarcoma patients remains to be defined [1,2,42], pending the results of clinical trials or meta-analyses from multi-institutional data.

No specific radiological and gross features have been reported thus far. Tumors are often extensively hemorrhagic and necrotic. MRI studies of CIC rearranged sarcomas usually show tumor originating from the deep soft tissues of the trunk, pelvis, or proximal extremities with heterogeneous but intense post-contrast enhancement [2], as in the present case.

Microscopically, *CIC* rearranged sarcomas are characterized by a more heterogeneous appearance than usual ES. They more frequently show “atypical features”, including plasmacytoid and rhabdoid features in approximately 25% and spindling in about 10%. They also show greater nuclear pleomorphism [16,31,32,39,43] with scant cytoplasm and prominent nucleoli, the latter being a useful initial diagnostic clue [2,44]. Myxoid stromal changes mimicking high-grade extra-skeletal myxoid chondrosarcoma or myoepithelial carcinoma are reported in some case [25,31]. They completely lack evidence of neuroectodermal differentiation, as Homer Wright rosettes that can be present in ES [2]. Mitotic rates are high, with almost all cases showing >10 mitotic figures per 10 high power fields [2,31]. Areas of necrosis are commonly observed (Table 1) [31,45].

Immunohistochemically, *CIC* rearranged sarcomas have been reported to show a unique expression pattern from ES (see Table 1 readapted from [2]). Up to 75% of cases co-express membrane CD99, usually focal and patchy, unlike the diffuse pattern observed in classical ES, and nuclear WT1 [2,17,22,46]. The vast majority of *CIC–DUX4* sarcomas express ERG and FLI1, as frequently as in classic ES [22], representing an important diagnostic challenge in differential diagnosis. *CIC* rearranged sarcomas often express diffuse nuclear ETV4, [46,47]. On the contrary, ES, *BCOR* rearranged sarcoma, and poorly differentiated synovial sarcomas, are negative for both nuclear ETV4 and WT1 staining [2,46]. Of note, nuclear WT1 positivity is also found in desmoplastic small round cell tumor and Wilms tumor, which should be considered in morphologic differential diagnosis [2]. The downstream target of *EWSR1-FLI1*, *NKX2-2*, is expressed in virtually all ES but is negative in *CIC* rearranged sarcomas [2,25,48]. Diffuse MYC immunohistochemical staining has been shown to be highly specific, detectable in 10 out of 10 cases of *CIC-DUX* sarcomas [45]. *CIC* rearranged sarcomas could also have positivity for calretinin, thus overlapping with mesothelioma [25]. A small number of cases can express focal desmin and cytokeratin AE1/AE3 (approximately 15% and 4%, respectively) [31].

Recently, DUX4 immunostaining has been reported to be diffusely positive in all cases of *CIC* rearranged sarcomas evaluated (5/5 cases), but negative among other SBRCTs, including ES [49]. The present case was characterized by weakly focal CD99 immunoreactivity, and diffuse nuclear positivity for WT1, FLI1, and TLE1. Both morphology and immunohistochemistry (IHC) features prompted molecular testing, necessary to confirm the diagnosis [22].

However, as for our case, decision making might be complicated by false negative results in molecular testing of such tumors [12,16,31,37]. Recently, an overall low performance of break-apart FISH and total RNA sequencing as molecular tests for SBRCTs with *CIC* rearrangement was reported [12,25,50]. Indeed, complex alterations involving the *CIC* locus on chromosome 19q, may have an adverse effect on FISH analysis [12]. Furthermore, automated algorithms for fusion discovery from RNA-Seq data could not detect the underlying gene fusion due to the highly repetitive DNA sequences juxtaposing the breakpoint in *CIC* [12]. In these cases, the underlying *CIC–DUX4* fusion could only be detected by manually reviewing the reads of these genes [12]. While the revision of this manuscript was ongoing, we had the opportunity to further analyze the case by RNA-Seq. Finally, we detected the CIC-DUX4 rearrangement, thus confirming the results of the methylation profiling. In our case, RNA-Seq was able to detect the translocation but in the experience reported by Koelsche and colleagues it failed in three out of six CIC rearranged tumors [12].

Use of a surrogate marker could be a more viable approach for facilitating the diagnosis of SBRCTs. SBRCTs with *CIC–DUX4* fusion have been shown to upregulate ETS transcription factors (*ETV1* on chromosome 7p, *ETV4* on chromosome 17q, *ETV5* on chromosome 3q) [39,43,47,50]. RNA in situ hybridization performed on formalin-fixed paraffin-embedded tissues to detect *ETV1*, *ETV4*, and *ETV5* transcripts was reported as a specific and sensitive ancillary test in cases with limited sample availability [39,51]. Recently, *PAX7* was identified as highly differentially expressed between ES and *CIC* rearranged sarcoma, also at the protein level. Indeed, Charville et al. demonstrated that PAX7 expression evaluated by IHC was detectable in 102/103 ES and 0/27 *CIC* rearranged sarcomas [52].

Accurate pathologic diagnosis is essential to make optimal therapeutic choices for patients with cancer, avoiding both over- and under-treatment and allowing appropriate inclusion of patients in clinical trials. In recent years, genome-wide DNA methylation array analysis has allowed identification and characterization of episignatures for an increasing number of diseases [53,54].

DNA methylation profiling has been shown to be a robust and reproducible approach for the classification of several tumor entities across age groups [12,18,21,38,54]. It takes advantage of the concept that the epigenetic signature in cancer is a combination of both somatically acquired DNA methylation changes and features that reflect the tumor cell origin in a lineage-dependent manner [18,21,55]. Epigenetic machinery regulates gene functions during developmental programs, tightly defined in space and time.

DNA methylation and histone modifications are among the most widely characterized epigenetic mechanisms of interest in these cases. CpG islands are variably distributed in the human genome; they are often located in gene promoters and act as key regulators of gene expression [56]. DNA methyltransferases catalyzes the covalent modification of the cytosine bases in CpG islands to generate 5-methylcytosine. This methylation makes DNA inaccessible to transcription factors and can also drive the formation of heterochromatin by histone modifiers.

In cancer, alteration in DNA methylation patterns can both transcriptionally inactivate tumor suppressors and increase oncogenes expression, sustaining tumorigenesis. Aberration in epigenetics can alter normal developmental programs and cell differentiation, leading to an uncontrolled proliferative state of progenitor cells [56]. Cytosine methylation at single-nucleotide resolution can be easily achieved by taking advantage of DNA bisulfite treatment. Such chemical process converts cytosine to uracil while not affecting 5-methylcytosine.

Array-based methods have progressively improved to reach over 850,000 CpG analyses throughout the human genome, such as the Illumina Infinium Human BeadChip arrays (450K and EPIC) that we used for our analysis [21,56]. These arrays have been shown to be rapid and cost-effective methods for genome-wide coverage of methylation patterns, providing critical insights into the epigenetic landscape of cancer [21,56]. They work comparably well on both formalin-fixed paraffin-embedded (FFPE) and frozen tumor samples from a relatively low DNA input (around 250 nanograms). Of note, these techniques show a rather short execution and analysis time (within a week), an appropriate timeframe for diagnostic implementation and use in the clinic.

Paralleling this technical progress, computational methods have been developed to deal the raw data in biologically and clinically applicable output [12,21,56]. These bioinformatics pipelines and classifiers use statistical tests and algorithms, such as random forest analyses, to make class predictions, exploiting advanced machine learning techniques. The main factor for the usefulness of DNA methylation profiling lies in the classification and clinical management of central nervous system tumors [19,21].

This approach also has been shown to be useful for addressing undifferentiated SBRCTs not otherwise specified [12]. Koelsche and colleagues identified 30 tumors failing to exhibit the typical ES translocation in a cohort of more than 1000 tumors considered as Ewing sarcomas. The authors performed tumor methylation profiling and analyzed data by unsupervised clustering and t-distributed stochastic neighbor embedding analysis, comparing with a reference methylation data set of 460 well-characterized prototypical sarcomas containing 18 subtypes. Tumors were assigned to Ewing sarcoma in 14 (47%), to SBRCTs with *CIC* alteration in 6 (20%), to SBRCTs with *BCOR* alterations in 4 (13%), to synovial sarcoma and to malignant rhabdoid tumors in 2 cases each, 1 case to mesenchymal chondrosarcoma, and 1 to adamantinoma. For cases assigned to the SBRCTs with *CIC* alterations, methylation class was verified by additional FISH and total RNA sequencing. Four of the six tumors revealed a *CIC* break-apart signal in the FISH analysis, indicating a rearrangement of the *CIC* locus. In two of them, RNA sequencing analysis detected a *CIC–DUX4* fusion. RNA sequencing also revealed a *CIC–DUX4* fusion in one of the two cases without a *CIC* break-apart signal [12].

This breakthrough tool has allowed us to successfully address a challenging diagnosis. Overall, DNA methylation profiling could have clinical implications in the molecular diagnostics of SBRCT, as already documented in central nervous system tumors [21,38].

## 4. Conclusion

*CIC* rearranged sarcomas are rare tumors that present substantial diagnostic challenges; morphological diagnosis is difficult. Since the immunohistochemical expression pattern seen in these tumors is unique, this is a useful adjunct in orienting a pathologic definition. Molecular testing is necessary to confirm diagnosis, but FISH and total RNA sequencing may sometimes have pitfalls.

In our view, DNA methylation profiles represent a highly useful instrument in the diagnosis and the molecular characterization of SBRCTs, with potential application in almost all pathology fields.

## Figures and Tables

**Figure 1 ijms-21-01818-f001:**
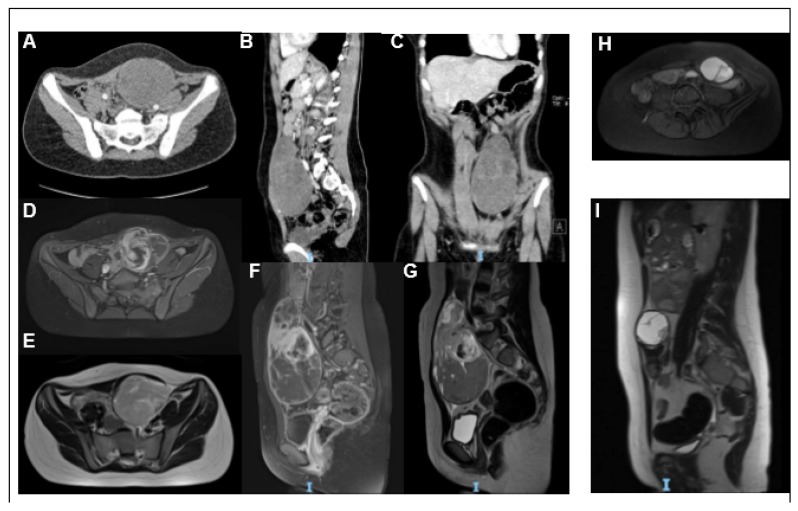
Imaging: Axial (**A**), sagittal (**B**), and coronal (**C**) Computed tomography (CT) scan images showing a gross expansive non-infiltrating mass, with clear and well-defined margins and an oval and oblong appearance. In coronal sections, it clearly shows a bilobated aspect with a homogeneously hypodense apical portion and a well-capped lower portion, which has a more uneven density, as in fluid internal lacunae. The lesion has a maximum size of 81.9 mm × 77 mm and a deep cranium-caudal extension of approximately 14 cm, up to the pelvic floor and posterior to the lumbosacral spine, which appears to have a cleavage plane. Magnetic resonance imaging (MRI) images (**D**–**I**) of the expansive process at diagnosis (**D**–**G**) and after neoadjuvant chemotherapy (**H**–**I**). T1w-fat sat axial (**D**) and sagittal (**F**), T2w axial **(E**) and sagittal (**G**) images show a heterogeneous mass extending in the left abdominal wall, in the left paramedian seat at the level of the navel. T1w-fat sat axial images (**H**) and T2w sagittal (**I**) images show a clear reduction in the dimensions of the known expansive formation after chemotherapy.

**Figure 2 ijms-21-01818-f002:**
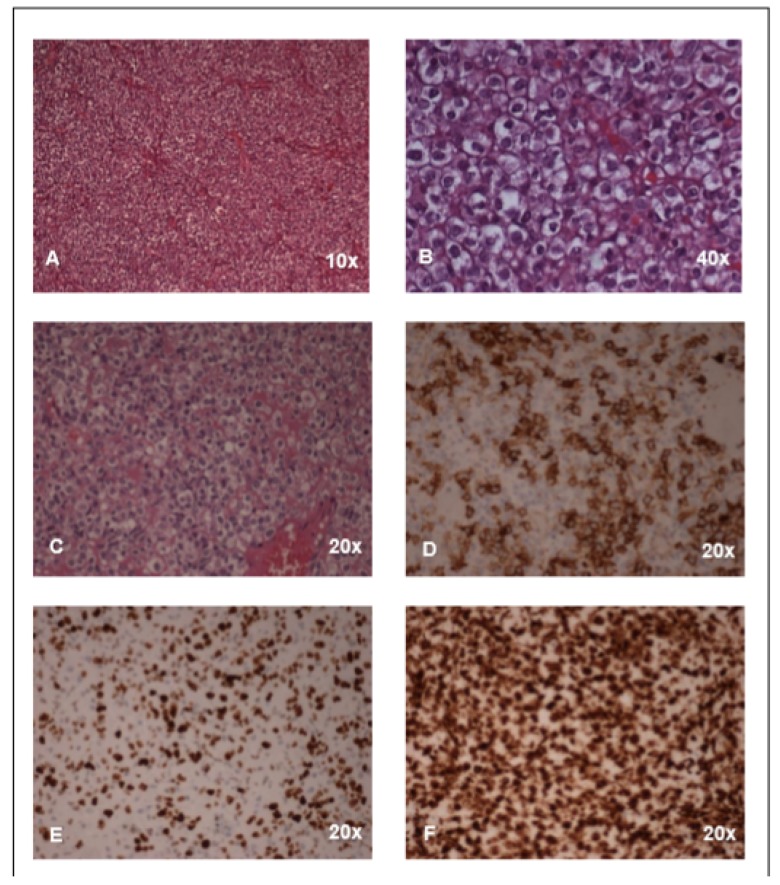
Histology: (**A**–**C**) Representative images (hematoxylin/eosin staining) of the malignant undifferentiated mesenchymal neoplasia. The cells were mostly epithelioid with clear cytoplasm in a solid pattern. (**D**–**F**) Immunostaining for CD99 (**D**), Ki67 (**E**), and WT1 (**F**).

**Figure 3 ijms-21-01818-f003:**
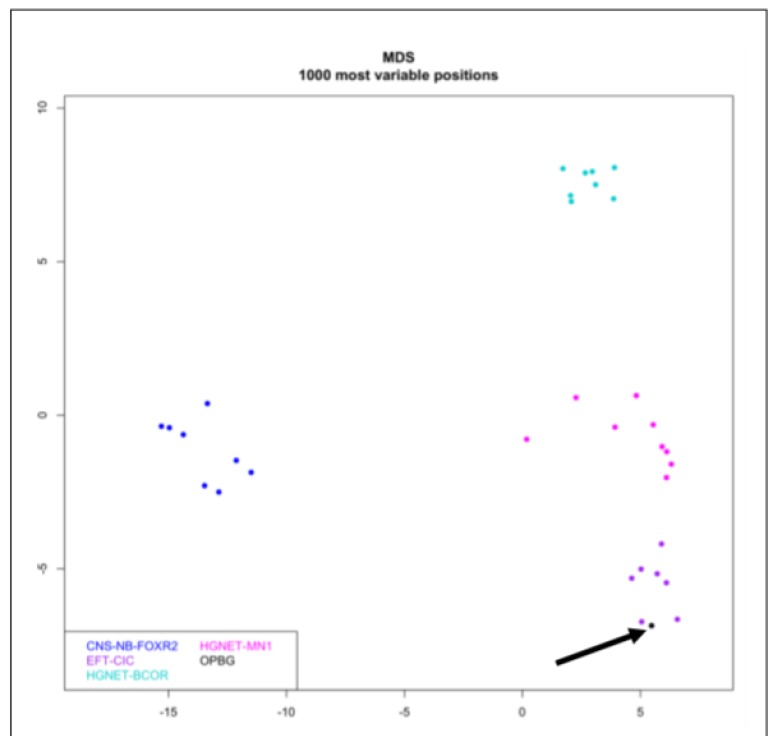
Multidimensional scaling (MDS) analysis performed on the 1000 most variable probes of the whole genome DNA methylation data shows a close similarity between the present case (OPBG) and *CIC* rearranged sarcomas, while clearly separate from other tumor entities. Color legend of the MDS plot as follows: *CIC* rearranged sarcoma case (black and arrowed); central nervous system neuroblastoma *FOXR2* (*CNS-NB-FOXR2*) (blue); EFT*-CIC* (violet); high-grade neuroepithelial tumor-*MN1* (HGNET*-MN1*) (pink); high-grade neuroepithelial tumor *BCOR* (HGNET*-BCOR*) (cyan).

**Figure 4 ijms-21-01818-f004:**
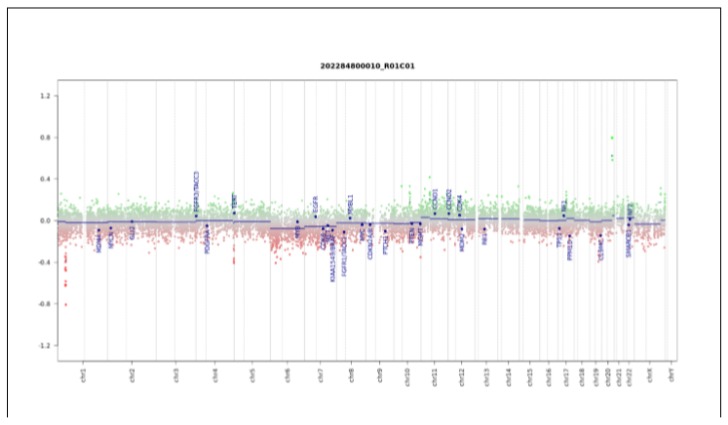
Copy number variation (CNC) profile analysis. Depiction of structural rearrangements involving autosomes and X/Y chromosome. Gains/amplifications represent positive (green) and losses represent negative (red) deviations from the baseline. Twenty-nine tumor relevant genomic regions are highlighted.

**Table 1 ijms-21-01818-t001:** Differential diagnosis of round cell sarcomas (readapted [2]).

	CIC-Rearranged Sarcoma	BCOR-Rearranged Sarcoma	Ewing Sarcoma	Poorly Differentiated Synovial Sarcoma	Desmoplastic Small Round Cell Tumor	Alveolar Rhabdomyosarcoma	Mesenchymal Chondrosarcoma	High-Grade Myxoid Liposarcoma
Peak incidence	3rd–4th decade	2nd decade	2nd decade	4th decade	3rd decade	2nd decade	3rd decade	4th decade
Most common anatomic site	Soft tissues of trunk/pelvis and extremities	Pelvic bones and metadiaphysis of long bones of lower extremities	Metadiaphysis of long bones, ribs, pelvis	Soft tissues of extremities	Soft tissues and viscera of abdomen	Soft tissues of extremities	Facial bones, long bones of lower extremities, and pelvic bones	Soft tissues of lower limbs, especially thigh
Cytomorphology	Round cells with moderate eosinophilic cytoplasm		Monotonous round cells	At least focal spindling	Monotonous round cells, minimal cytoplasm	Monotonous round cells, rare multinucleated giant cells or striated cells	Round cells and spindled cells	Round cells with rare univacuolated lipoblasts
Nuclear features	Vesicular chromatin, prominent nucleoli, moderate pleomorphism	Fine chromatin, inconspicuous nucleoli	Fine chromatin, inconspicuous nucleoli	Vesicular chromatin, prominent nucleoli	Uniform hyperchromasia, inconspicuous nucleoli, and minimal eosinophilic cytoplasm	Large, fine chromatin, prominent nucleoli	Small, fine chromatin, inconspicuous nucleoli	Vesicular chromatin, prominent nucleoli, moderate pleomorphism
Stroma	Myxoid changes common, subset with dense hyaline bands	Subset with myxoid changes	Scant	Scant	Abundant desmoplastic stroma encasing round cell nests	Fibrotic septae between anastomosing round cells	Geographically variable chondroid matrix	Myxoid changes
Typical positive immunohistochemical profile	Patchy CD99, ETV4, WT1, can be ERG and FLI1	Patchy CD99, TLE1, BCOR, CCNB3	Diffuse membranous CD99, NKX2-2	TLE1, patchy EMA and keratin	WT1, keratin, EMA, desmin (perinuclear)	Desmin, myogenin, MyoD1	Diffuse membranous CD99, NKX2-2, S100 in chondroid areas	S100
Genetics (in order of incidence)	CIC-DUX4, CIC-DUX4L, CIC-FOX04	BCOR-CCNB3, BCOR-MAML3, ZC3H7B-BCOR	EWSR1-FLI1, EWSR1-ERG, EWSR1-PATZ1, also FUS-ERG	SS18-SSX1, SS18-SSX2	EWSR1-WT1	PAX3-FOX01, PAX7-FOXO1	HEY1-NCOA2	FUS-DDIT3 EWSR1-DDIT3

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
