# Peer review of "DNA Methylation Profiling for Diagnosing Undifferentiated Sarcoma with Capicua Transcriptional Receptor (CIC) Alterations"

_ijms, 2020, doi:10.3390/ijms21051818_

Round 1

Reviewer 1 Report

Summary: Here, the authors discuss the use of DNA-methylation analysis for the proper diagnosis of undifferentiated pediatric soft tissue sarcomas. In the review, they describe the case of an undifferentiated sarcoma of the abdominal wall in a 12-year-old girl, and how they used DNA-methylation profiling to classify the neoplasia as small blue round cell tumor with capicua transcriptional receptor (CIC) rearrangements.

Major Comments:

It is unusual for a case report to be included in a review article. Was this meant to be submitted as a review? This article would be better separated into a case report with a proper materials and methods section, and then a review article summarizing your findings and the findings of others.

Author Response

Reviewer 1

Comments and Suggestions for Authors

Summary: Here, the authors discuss the use of DNA-methylation analysis for the proper diagnosis of undifferentiated pediatric soft tissue sarcomas. In the review, they describe the case of an undifferentiated sarcoma of the abdominal wall in a 12-year-old girl, and how they used DNA-methylation profiling to classify the neoplasia as small blue round cell tumor with capicua transcriptional receptor (CIC) rearrangements.

Major Comments:

It is unusual for a case report to be included in a review article. Was this meant to be submitted as a review? This article would be better separated into a case report with a proper materials and methods section, and then a review article summarizing your findings and the findings of others.

Authors’ reply: we took the cue from our experience to widely discuss the available literature in the field of CIC- rearranged sarcomas, as well as the use of DNA methylation profiling in the diagnosis of pediatric tumors.

The decision to include the case report to introduce the literature review has been taken in accordance with the Editorial board. The article has been reorganized and restructured to make it more comprehensive and readable for the journal audience.

We firmly believe in the novelty and the scientific message of this article.

Reviewer 2 Report

The manuscript entitled “DNA methylation profiling for diagnosing undifferentiated sarcoma with Capicua transcriptional receptor (CIC) alterations” by Miele et al. discussed the possibility of DNA methylation profile to classify a diagnostically challenging tumor.  They have reported that they successfully treated a patient with a drug that was selected based on the DNA methylation profile. The authors also discussed the classification of undifferentiated sarcoma.

Broad comments: While this article is supposed to be a review article, the primary discussion point of the current manuscript is not clear. As a result, references are not well focused, and discussion is not deep enough as a review article. Although the successful treatment is an excellent achievement for the patient and the team, classifying tumors based on DNA methylation is not novel.  

There are several interesting points they tried to discuss. Such as while they did not detect molecular alterations in the patient including CIC-DUX4, the tumor was classified as CIC alteration ES. Moreover, the treatment (neoadjuvant chemotherapy), which worked on the patient, has minimal effect on most of CIC rearranged sarcoma patients. However, the authors did not discuss why or how these discrepancies.

Author Response

Reviewer 2

Comments and Suggestions for Authors

The manuscript entitled “DNA methylation profiling for diagnosing undifferentiated sarcoma with Capicua transcriptional receptor (CIC) alterations” by Miele et al. discussed the possibility of DNA methylation profile to classify a diagnostically challenging tumor.  They have reported that they successfully treated a patient with a drug that was selected based on the DNA methylation profile. The authors also discussed the classification of undifferentiated sarcoma.

Broad comments: While this article is supposed to be a review article, the primary discussion point of the current manuscript is not clear. As a result, references are not well focused, and discussion is not deep enough as a review article. Although the successful treatment is an excellent achievement for the patient and the team, classifying tumors based on DNA methylation is not novel. 

Authors’ reply: We thank the reviewer for carefully reading our manuscript. We have accurately revised our manuscript accordingly. A point-to-point response to the reviewers’ comments, which are reported in blue, is given below.

The novelty of our experience was to “classify” an unclassifiable disease with current molecular biology techniques (FISH and RT-PCR) by the use of methylation profiling. We compared the methylation data of our patient with those of available CIC-rearranged tumor. Of note, DNA methylation profiling for classifying tumors is an expanding tool that has been well established in brain tumors (Capper D. et al. 2018). However few researches have been published in the field of sarcomas, and very few cases of CIC-rearranged sarcomas have been reported (6 cases of which only 1 aged <20 years in Koelsche C. et al, 12 cases of CNS-EFT-CIC-in Sturm D et al 2016; 13 CNS-EFT-CIC-in Capper D et al 2018). Thus, we firmly believe in the novelty and the scientific message of this article.

There are several interesting points they tried to discuss. Such as while they did not detect molecular alterations in the patient including CIC-DUX4, the tumor was classified as CIC alteration ES.

Authors’ reply:  we thank the reviewer for raising this point. While the revision of this manuscript was ongoing, we further analyzed the case by Next Generation Sequencing (Archer technology) and finally detected the CIC-DUX4 rearrangement (chr19:42799214,chr4:191006179), confirming the results obtained by methylation profiling. We added this point and discussed in the text the results and the importance of this novel tool to help in the diagnosis of such rare tumors.

Moreover, the treatment (neoadjuvant chemotherapy), which worked on the patient, has minimal effect on most of CIC rearranged sarcoma patients. However, the authors did not discuss why or how these discrepancies.

Authors’ reply: We thank again the reviewer for raising this point. CIC-rearranged sarcomas are very rare tumor entity, even rarer in the pediatric age, and their diagnosis is very challenging. The most of the data available on chemotherapy response in CIC rearranged sarcoma patients refer to older population, so the discrepancy between our patient response and the literature report could be found in the young age of our patient. We discuss this issue in the text.

Reviewer 3 Report

This study is a valuable molecular case report and review of the literature which applies DNA methylation array profiling to diagnose and treat a patient with CIC-rearranged sarcoma. The discussion section provides a comprehensive and excellent review of this disease entity. I have the following suggestions for revision prior to publication of this manuscript.

The abstract and case report sections of the manuscript are very wordy and contain too many commas. The sentence structure is unnecessarily complex and is difficult for the reader to follow. I suggest that these sections be rewritten with an emphasis on simplifying the sentence structure and reducing the number of commas. This section would benefit from English language scientific editing. Interestingly, the discussion section does not seem to contain these shortcomings. Perhaps the authors can remedy this discrepancy. The font size changes at several locations in the manuscript (from line 49-50, 320-324, and 378-379) Please make sure this is uniform and meets the journal format. Figure 1 – the label for part H is missing On line 150 it states the “margins of resection fell into healthy tissue.” This statement is confusing. Were the margins of resection positive or negative? Figure 3 – Draw an arrow to the CIC rearranged sarcoma case. The black dot is difficult to distinguish from the surrounding purple dots. I think this would be helpful to the reader. Please describe what happened to the pulmonary nodules with treatment. Did they disappear, shrink, grow? The information in line 308-314 is critically important to understand the rationale for using methylation analysis to diagnosis CIC-rearranged sarcomas. The authors state that RNA-seq is not capable of reliably identifying some CIC-rearrangements due to highly repetitive sequences juxtaposing the breakpoint. This needs to be set up in the introduction section of the paper to make sure the reader understands the underlying rationale. The paragraph from line 322-378 is too long and difficult to digest. Please break up into multiple paragraphs. In lines 101-103, please clarify what is meant by “discussed with central histolopathological reference panel.”

Author Response

Reviewer 3

This study is a valuable molecular case report and review of the literature which applies DNA methylation array profiling to diagnose and treat a patient with CIC-rearranged sarcoma. The discussion section provides a comprehensive and excellent review of this disease entity. I have the following suggestions for revision prior to publication of this manuscript.

Authors' reply: We greatly appreciate the Reviewer’s positive comments. We have carefully revised the manuscript according to the Reviewer’s insightful comments and provided point-by-point responses as follows.

The abstract and case report sections of the manuscript are very wordy and contain too many commas. The sentence structure is unnecessarily complex and is difficult for the reader to follow. I suggest that these sections be rewritten with an emphasis on simplifying the sentence structure and reducing the number of commas. This section would benefit from English language scientific editing.

Authors' reply: the article was extensive reviewed for editing.

Interestingly, the discussion section does not seem to contain these shortcomings. Perhaps the authors can remedy this discrepancy. The font size changes at several locations in the manuscript (from line 49-50, 320 324, and 378-379)

Authors' reply: we corrected the font size.

Please make sure this is uniform and meets the journal format. Figure 1 – the label for part H is missing.

Authors' reply: we reformatted the Figure 1 and inserted the missed “H” label. We uploaded the figures in a separate file from the text.

On line 150 it states the “margins of resection fell into healthy tissue.” This statement is confusing. Were the margins of resection positive or negative?

Authors' reply: we rephrased the sentence “Margins of resection were negative”.

Figure 3 – Draw an arrow to the CIC rearranged sarcoma case. The black dot is difficult to distinguish from the surrounding purple dots. I think this would be helpful to the reader.

Authors reply: As suggested, we drown an arrow to better indicate our CIC-rearranged sarcoma case.

Please describe what happened to the pulmonary nodules with treatment. Did they disappear, shrink, grow?

Authors' reply: We thank the reviewer for raising this issue. The pulmonary nodules remained stable in dimension and features after treatment. We described this finding in the text.

The information in line 308-314 is critically important to understand the rationale for using methylation analysis to diagnosis CIC-rearranged sarcomas. The authors state that RNA-seq is not capable of reliably identifying some CIC-rearrangements due to highly repetitive sequences juxtaposing the breakpoint. This needs to be set up in the introduction section of the paper to make sure the reader understands the underlying rationale.

Authors' reply: as pointed out by the reviewer, the novelty of our experience was to “classify” an unclassifiable disease with current molecular biology techniques (FISH and RT-PCR) by the use of methylation profiling. We compared the methylation data of our patient with those of available CIC-rearranged tumor. While the revision of this manuscript was ongoing, we further analyzed the case by Next Generation Sequencing (Archer technology) and finally detected the CIC-DUX4 rearrangement, confirming the results obtained by methylation profiling. In our case, RNA-Seq was able to detect the translocation but in the experience reported by Koelsche and colleagues it failed in 3 out of 6 CIC-rearranged tumors. We added this point and discussed in the text the results and the importance of this novel tool to help in the diagnosis of such rare tumors. As suggested, to stress the rationale for using methylation analysis to diagnosis CIC-rearranged sarcomas, we added the following sentence in the introduction section: “However, FISH and total RNA sequencing may sometimes have pitfalls in identifying some rearrangements." 

The paragraph from line 322-378 is too long and difficult to digest. Please break up into multiple paragraphs.

Authors' reply: as suggested, the article was extensive reviewed for editing.

In lines 101-103, please clarify what is meant by “discussed with central histolopathological reference panel.”

Authors reply: we rephrased the sentence, as it follows  “Histopathological diagnosis was determined through examination of the specimen at the Italian Pediatric Sarcoma central pathology panel”.

Round 2

Reviewer 2 Report

Most of my concerns were addressed.

Reviewer 3 Report

All of my concerns have been adequately addressed in this revised version and I recommend the manuscript be published without additional revision.